# Inputs and Outputs of the Mammalian Circadian Clock

**DOI:** 10.3390/biology12040508

**Published:** 2023-03-28

**Authors:** Ashley N. Starnes, Jeff R. Jones

**Affiliations:** Department of Biology, Texas A&M University, College Station, TX 77843, USA

**Keywords:** circadian, suprachiasmatic, circuits

## Abstract

**Simple Summary:**

In mammals, circadian rhythms in nearly all behaviors and physiological processes are controlled by the suprachiasmatic nucleus, a collection of neurons in the hypothalamus that synchronize the brain and body to local time. However, the circuits linking the suprachiasmatic nucleus to both the external world and to downstream targets are poorly understood. This review describes the inputs to and outputs from the suprachiasmatic nucleus. A better understanding of the circadian “connectome” is essential to determine how the disruption of these circuits can negatively impact human health.

**Abstract:**

Circadian rhythms in mammals are coordinated by the central circadian pacemaker, the suprachiasmatic nucleus (SCN). Light and other environmental inputs change the timing of the SCN neural network oscillator, which, in turn, sends output signals that entrain daily behavioral and physiological rhythms. While much is known about the molecular, neuronal, and network properties of the SCN itself, the circuits linking the outside world to the SCN and the SCN to rhythmic outputs are understudied. In this article, we review our current understanding of the synaptic and non-synaptic inputs onto and outputs from the SCN. We propose that a more complete description of SCN connectivity is needed to better explain how rhythms in nearly all behaviors and physiological processes are generated and to determine how, mechanistically, these rhythms are disrupted by disease or lifestyle.

## 1. Introduction

As a consequence of evolving on a rotating planet, nearly all organisms exhibit circadian, or near-24-h, rhythms in behavior and physiology. These internally-driven rhythms can be mechanistically described in terms of an “eskinogram”, named after the chronobiologist Arnold Eskin. According to this heuristic, an input–most often light from the sun–changes the phase and/or period of an oscillator which, in turn, changes the oscillator’s outputs [1]. In mammals, the central circadian oscillator and pacemaker is the suprachiasmatic nucleus (SCN). The SCN comprises ~20,000 heterogenous, mostly gamma-aminobutyric acid (GABA)-ergic, neurons located at the base of the hypothalamus that are necessary for circadian rhythmicity [2,3,4]. In common with most, if not all, mammalian cells, each SCN neuron contains a “molecular clock”, a transcriptional/translational delayed negative feedback loop in which clock genes and proteins are transcribed, translated, and degraded once each day. SCN neurons also exhibit endogenous daily rhythms in electrical activity such that they fire action potentials more quickly during the day than at night even in the complete absence of synaptic drive. These molecular and electrical rhythms are synchronized across the SCN neural network through complex intercellular coupling mechanisms mediated by GABA and various neuropeptides including vasoactive intestinal peptide (VIP), arginine vasopressin (AVP), and gastrin-releasing peptide (GRP) [5,6]. In a typical eskinogram, the input onto the SCN network oscillator is usually modeled as light information via the retina and the output from the SCN is nebulously defined as daily rhythms in behavior and physiology. This model describes the canonical pathway of photoentrainment, wherein local time (environmental light) is encoded by molecular clocks within SCN neurons that propagate timing information onward to entrain molecular clocks in downstream cells and tissues [7,8] (Figure 1). However, photoentrainment is only one of the many SCN input-output pathways that together determine the body’s internal circadian time. In this review, we highlight what is currently known and unknown about the inputs onto and outputs from the SCN. We propose that fully understanding SCN connectivity using modern genetic tools will advance the eskinogram from a simplified conceptual model to one that positions the SCN in the center of a dynamic circadian network that incorporates bidirectional communication to and from clocks throughout the brain and body.

## 2. Inputs to the SCN: Afferent Projections

Perhaps unsurprisingly, the main afferent projection to the SCN originates in the retina. Melanopsin-expressing intrinsically photosensitive retinal ganglion cells (ipRGCs) project through the retinohypothalamic tract (RHT) to the SCN and other non-image forming visual brain regions such as the olivary pretectal nucleus (OPN) and intergeniculate leaflet (IGL) [28,29,30,31]. ipRGCs were historically thought to predominantly synapse on VIP- and GRP-producing neurons in the ventromedial SCN core because these neurons are preferentially activated by light [32,33]. However, individual ipRGCs have been shown to bilaterally innervate neurons throughout both the core and dorsolateral shell of the SCN, including those that produce AVP [34]. Light pulses that “phase shift”, or reset the timing of, circadian rhythms cause ipRGCs to co-release the excitatory neurotransmitter glutamate and, to a lesser extent, the modulatory neuropeptide pituitary adenylate cyclase-activating polypeptide (PACAP) onto SCN neurons [9,10]. Surprisingly, a subset of ipRGCs release GABA onto the SCN to dampen the sensitivity of the circadian system to dim light [35]. The threshold needed for ipRGC-mediated signals to activate neurons in the SCN is presumably higher for neurons in the dorsolateral SCN versus neurons in the ventromedial SCN, which may be reflected in the relative spatial distribution of glutamate and PACAP receptors [36]. Activating glutamate and PACAP receptors both directly and indirectly increases calcium influx into core SCN neurons through the opening of calcium-permeable N-methyl-D-aspartate (NMDA) glutamate receptors and by increasing firing rate, respectively. This response is then transmitted both intracellularly to the molecular clock itself through the activation of calcium-dependent kinases and intercellularly to other SCN neurons through the release of VIP or GABA [37,38]. ipRGC projections to the SCN through the RHT that were first identified in nocturnal rodents are largely conserved in diurnal animals [39]. However, while in nocturnal rodents the vast majority of SCN neurons are excited by light, in diurnal rodents, many SCN neurons are inhibited by light. This suggests that the ratio of GABA to glutamate released from ipRGCs onto the SCN is much greater in nocturnal rodents than in diurnal rodents [35,40,41]. The functional significance of these inhibitory projections in diurnal rodents is unknown but may mechanistically explain differences in photoentrainment between species.

Several non-retinal inputs to the SCN of nocturnal animals have also been identified using “conventional” retrograde tract-tracing methods (in contrast to more modern viral vector-based and genetically-encoded tracing methods; for review, see [42]). These include so-called “non-photic” projections from the median raphe nucleus in the brainstem and projections from the IGL in the thalamus via the geniculohypothalamic tract that convey both photic and non-photic information [43,44]. IGL neurons release GABA and the neuropeptides neuropeptide Y (NPY), enkephalin, and neurotensin onto neurons in the ventral SCN to reset the clock in response to light and, potentially, arousal stimuli [45]. Conversely, median raphe nucleus neurons exclusively release serotonin onto the ventral SCN to reset the clock in response to behavioral arousal and feedback from locomotor activity [46]. Conventional tracing methods have confirmed that, as in nocturnal rodents, the SCN of diurnal rodents receive NPYergic projections from the IGL and serotonergic projections from the median raphe nucleus [47,48,49]. Curiously, the SCN of the diurnal tree shrew (which is closely related to primates) also receives monosynaptic projections from unique non-photic brain regions including the locus coeruleus and periaqueductal gray [50]. 

More recently, viral vectors have been used to identify monosynaptic inputs onto specific genetically-defined subtypes of SCN neurons (Figure 2). VIP neurons receive projections from numerous brain regions in addition to the IGL and raphe nucleus, including the ventromedial hypothalamus (VMH), arcuate nucleus (ARC), medial preoptic area (MPOA), and paraventricular thalamus (PVT), and paraventricular nucleus of the hypothalamus (PVN) [51,52]. This widespread innervation is consistent with several conventional tracing studies that found that while the SCN is predominantly innervated by the retina, IGL, and raphe nucleus, around 40 different brain regions project monosynaptically to the SCN [53,54]. Surprisingly, SCN neurons that produce the neuropeptide cholecystokinin (CCK) receive monosynaptic inputs from the ARC, VMH, and PVN, but do not receive inputs from the primary SCN-projecting structures (retina, IGL, or raphe nuclei) [55]. Accordingly, SCN CCK neurons do not respond to light and do not contribute to photoentrainment [56]. It is unclear what information is being conveyed to CCK, VIP, and other SCN neurons by these “secondary” projections but it may include information about arousal state and motivation [57,58,59,60]. SCN GRP, but not SCN AVP, neurons appear to receive projections from the IGL, dorsal raphe nucleus, and median raphe nucleus [6,61]; however, these were discovered using conventional tracing in combination with immunohistochemistry. Dopamine receptor 1a-expressing SCN neurons receive sparse monosynaptic projections from dopaminergic neurons of the ventral tegmental area (VTA). The release of dopamine onto the SCN has been functionally linked to photoentrainment and, intriguingly, weight gain associated with hedonic feeding [62,63]. However, the origins and functions of additional monosynaptic projections to these and other SCN subpopulations have not yet been determined.

## 3. Inputs to the SCN: Hormones

SCN neurons integrate these diverse synaptic inputs from the brain and retina with hormone receptor activation by circulating endocrine cues from the periphery [64]. Most research on hormonal signaling to the SCN has focused on steroid hormones that can easily cross the blood-brain barrier. These include aromatizable (e.g., testosterone) and non-aromatizable (e.g., dihydrotestosterone) androgens, which bind the androgen receptor (AR), and estrogens (e.g., estradiol), which bind estrogen receptors (ER) alpha and beta [65,66,67,68,69]. Aromatizable androgens can also be aromatized into estrogens that subsequently interact with ERs (see below). ARs are largely located in the ventral SCN core where they colocalize with retinorecipient GRP-producing neurons (~85%) and, to a much lesser extent, VIP neurons (~12%). ARs are essentially absent from AVP neurons in the SCN shell (<2%) [70]. AR expression levels in the SCN of male rodents and humans are much greater than levels in the SCN of females. In rodents, this sexual dimorphism is dependent on androgens, as castration reduces, and androgen replacement dose-dependently restores, SCN AR expression levels in male mice [71]. The physiological effect of SCN AR activation is complex. ARs normally interact with androgen response elements on several core clock genes (including the negative regulator *Per1*) to dose-dependently alter transcriptional activity [70,72]. ARs also modulate the responsiveness of SCN neurons to light. Consequently, in dim light, androgen removal reversibly lengthens the free-running period of locomotor activity rhythms but has no effect in constant darkness [65]. Exogenous testosterone also increases the period of PER2::LUC bioluminescent clock reporter rhythms in ex vivo SCN slices [73].

While all ARs are located almost exclusively in the ventral SCN, both ER isoforms are located predominantly on non-AVP neurons in the SCN shell. ERs are essentially absent from GRP and VIP neurons and, consequently, have very little spatial overlap with ARs. ERα expression in the SCN is sparse (~3% of neurons), but ERβ expression is much more robust and has pronounced sexual dimorphism (~15% of neurons in males, ~25% in females) [74]. Estrogen removal reversibly increases the free-running period of locomotor activity in several species [75,76]. This likely underlies the phase advance in locomotor activity onset (“scalloping”) observed every 4–5 days during proestrus in cycling female hamsters and rats but, notably, not in mice [75,77,78]. Several studies using breast cancer cell lines have shown that estrogen-ER complexes interact with estrogen response elements on the core clock genes *Per2* and *Clock* to increase their transcription rate [79,80]. Culturing mouse uterus in estrogen-containing media consequently shortens the period of PER2::LUC bioluminescence rhythms. This chronic estrogen treatment has no effect on the period of PER2::LUC rhythms in cultured SCN slices [81]. The effects of acute estrogen treatment on the phase (or “peak time”) of SCN rhythms are unknown. Acute estrogen treatment can, however, depolarize SCN neurons and dose-dependently increase their spontaneous firing rate [82]. Because increasing SCN firing rate can reset the phase of the molecular clock, this may be a mechanism by which estrogen influences the timing of SCN bioluminescence rhythms [83]. In support of this, *Per1-luc* bioluminescent clock reporter rhythms in the cultured SCN of ovariectomized rats peak much earlier in the day than rhythms in normally-cycling rats [84]. In general, other steroid hormone receptors are sparsely expressed in the SCN. The progesterone receptor is essentially absent in the SCN of rodents but is present in the SCN of humans (and, perhaps, in non-human primates) with no observed sexual dimorphism [85]. Progesterone normally interacts with estrogen to influence circadian locomotor behavior [86]. However, this interaction likely occurs downstream from the SCN as physiological levels of progesterone have no effect on SCN *Per1-luc* rhythms [84]. In adult rodents, the glucocorticoid receptor (GR) is ubiquitous throughout the brain and body with the notable exception of the SCN [87]. This indicates that the SCN could potentially control the circadian release of glucocorticoids to entrain GR-expressing cells and tissues in the periphery without feeding its timing information back onto itself [88,89,90]. Curiously, GR protein and mRNA are both highly expressed in the SCN of neonates, and several studies suggest that very low levels of GR in the SCN may persist into adulthood [91,92,93].

Although not a steroid hormone, melatonin readily crosses the blood brain barrier to feed back onto the SCN [94]. In mammals, melatonin acts on two high-affinity G-protein coupled receptors: MT1 and MT2. MT1 is localized to a small number of discrete brain regions including the PVT, pars tuberalis, and, notably, the SCN. Conversely, MT2 is broadly distributed across the brain [95]. MT1 is highly expressed throughout the SCN with a higher density in neurons in the ventrolateral core. MT2 is also expressed throughout the SCN, though at a much lower level [95]. The expression of melatonin receptors in the SCN results in a reciprocal relationship where the SCN both directs melatonin synthesis (see below) and receives melatonin feedback. This feedback, in turn, alters SCN clock gene expression and electrical activity [96,97]. Although MT1 and MT2 each inhibit cyclic AMP response element-binding protein (CREB) phosphorylation to regulate clock gene expression, the specific functions of melatonin feedback signals on the SCN are differentiated by receptor subtype [98]. MT1 receptor activation acutely inhibits SCN neuronal activity and PACAP-mediated signal transduction [99]. MT2 receptors instead mediate melatonin’s phase-shifting effect on SCN electrical activity which persists in mice genetically deficient for MT1 but is abolished by MT2-specific antagonists [100] (but see [101]). Likely due to the redundant nature of the circadian system, the magnitude of melatonin’s influence on circadian output is seemingly mild. Although exogenous melatonin can entrain circadian rhythms in SCN gene expression, SCN electrical activity, and circadian behavior, endogenous melatonin is not necessary for rhythmic behavior under entrained conditions [102,103,104,105]. Instead, melatonin (acting through MT1 in the absence of light) stabilizes behavioral rhythms by promoting uniformity in day-to-day activity patterning [106]. MT2 activation promotes faster entrainment to new light cycles as mice with intact MT2 exhibit accelerated entrainment compared to MT2-deficient mice [107]. Prenatally, melatonin acts to synchronize fetal and maternal circadian rhythms [108]. Mammalian fetuses cannot synthesize melatonin and must rely solely on maternal melatonin (and other factors; for review see [109]) as a time-of-day signal [110]. Maternally-derived melatonin readily moves through the placenta to enter the fetal circulation and bind to melatonin receptors in the fetal SCN [111]. The importance of melatonin in entraining fetal clocks is variable and likely organ- and species-specific. For example, maternal pinealectomy results in fetal arrhythmicity in sheep, but has no effect on fetal rhythms in rats [112,113]. Similarly, fetal rhythms persist in the widely-used melatonin-deficient C57BL/6J strain of laboratory mice [114]. Thus, at least in rodents, maternal melatonin is sufficient but not necessary to entrain the fetus.

## 4. Outputs from the SCN: Efferent Projections

Surprisingly, the SCN does not form widespread synaptic connections throughout the brain [6]. The major monosynaptic projections from the SCN are to a relatively small number of nuclei in the adjacent hypothalamus and thalamus (Figure 3a). These are mainly “periventricular” projections to hypothalamic nuclei adjacent to the third ventricle, including the subparaventricular zone (SPZ), paraventricular nucleus (PVN), and dorsomedial hypothalamus (DMH), and dorsal-rostral projections to the paraventricular nucleus of the thalamus (PVT) [115,116,117,118]. Anterograde tracing in Vip-IRES-Cre, Avp-IRES2-Cre, and Prok2-EGFP mouse lines has demonstrated that SCN subpopulations have characteristic projection patterns [51,119,120]. The DMH, SPZ, and PVT are each highly innervated by axons from VIP, AVP, and PK2-expressing SCN neurons. Axons from VIP neurons also project prominently to the anterior hypothalamus (AHN), ventromedial preoptic nucleus (VMPO), medial preoptic area (MPOA), paraventricular nucleus (PVN), periventricular nucleus (PeVN), lateral habenula (LHb), lateral hypothalamus (LH), and ventromedial hypothalamus (VMH). VIP axons project sparsely to the lateral septum, bed nucleus of the stria terminalis (BNST), ventrolateral preoptic nucleus (VLPO), median preoptic nucleus (MPN), anteroventral periventricular nucleus (AVPV), posterior hypothalamus (PH), and supraoptic nucleus (SON). Axons from AVP neurons project prominently to the VMPO, organum vasculosum of the lamina terminalis (OVLT), AVPV, PVN, and arcuate nucleus (ARC). AVP axons project sparsely to the lateral septum, AHN, BNST, VLPO, MPOA, MPN, PeVN, LHb, LH, VMH, PH, and SON. A recent study also identified monosynaptic projections from a small number of SCN AVP neurons to the central amygdala [121]. Axons from PK2 neurons project prominently to the lateral septum, MPOA, OVLT, PVN, LH, and periaqueductal gray. PK2 axons project sparsely to the BNST, LHb, ARC, dorsal raphe nucleus, supramammillary nucleus, and PH. Efferent projections from other subpopulations of SCN neurons (such as NMS+, GRP+, and CCK+) have not been fully characterized using modern genetic tools. However, intriguingly, SCN NMS axons project to a unique class of dopaminergic neurons in both the PeVN and PVN [122,123].

## 5. Outputs from the SCN: Physiological Rhythms

The function of most SCN projections to downstream brain regions is poorly understood. However, a few output circuits have been mapped from the SCN to rhythms in physiology and/or behavior (Figure 3b). For instance, SCN VIP neurons simultaneously activate gonadotropin-releasing hormone (GnRH)-producing neurons in the MPOA and inhibit gonadotropin inhibitory hormone (GnIH)-producing neurons in the DMH [124,125,126]. SCN AVP neurons activate kisspeptin-producing neurons in the AVPV which, in turn, activate GnRH neurons [127,128,129]. If this daily timing cue from the SCN is coincident with high levels of estrogen, GnRH neurons induce the gonadotrophs in the anterior pituitary to release a luteinizing hormone surge that triggers ovulation, which, in nocturnal rodents, occurs in the mid-afternoon once every 4–5 days [130]. The SCN also regulates body temperature rhythms through direct AVPergic projections to the MnPO and indirect projections to the ARC [131]. During the active phase (night in nocturnal rodents), the SCN activates ARC neurons that release α-melanocyte stimulating hormone (α-MSH) onto the MnPO to sustain high body temperature. During the inactive phase, the SCN simultaneously inhibits ARC α-MSH release and directly releases AVP onto the MnPO to lower body temperature. Because the ARC encodes metabolic information, thermoregulatory rhythms can be dynamically adjusted in response to an animal’s metabolic needs [132,133]. It is unclear how, mechanistically, the ARC integrates these circadian and metabolic cues.

The SCN also projects both directly and indirectly (through the SPZ) to the PVN to regulate endocrine and autonomic rhythms, including melatonin release, glucocorticoid release, and heart rate [27,134,135,136,137]. The multi-synaptic melatonin release circuit originates with SCN neurons that project to pre-autonomic PVN neurons that subsequently synapse on neurons in the intermediolateral column of the spinal cord (IML). IML neurons project to noradrenergic neurons in the superior cervical ganglion that stimulate the pineal gland to produce melatonin [138]. Sparse glutamatergic SCN projections provide a constant level of excitatory stimulation to the PVN. However, this weak signal is overwhelmed by GABAergic SCN projections that rhythmically inhibit PVN neurons (and, consequently, melatonin release) during the daytime in all vertebrates [135,138]. However, in both nocturnal and diurnal animals, the activity of corticotropin-releasing hormone (CRH)-producing PVN neurons promotes the release of adrenocorticotropic hormone (ACTH) from the pituitary to induce the adrenal glands to maximally release glucocorticoids just prior to waking (dusk or dawn, respectively). The adrenal glands also receive polysynaptic projections from pre-autonomic PVN neurons that gate their sensitivity to ACTH [139,140,141]. In nocturnal rodents, SCN AVP neurons inhibit PVN neurons either directly or indirectly through GABAergic interneurons in the SPZ [142]. In diurnal species, SCN AVP neurons are instead hypothesized to project to excitatory glutamatergic SPZ interneurons [143]. SCN VIP neurons similarly inhibit the PVN to suppress glucocorticoid release, likely through the paracrine release of VIP and/or the synaptic release of GABA [26,144,145]. A stimulatory signal from the SCN is thus far unidentified but may not be necessary to promote glucocorticoid release, as intrinsic PVN neuron rhythms may suffice to overcome weak inhibitory input from the SCN [144]. Inhibitory SCN projections to the PVN also regulate daily rhythms in autonomic function including, notably, heart rate [26]. SCN AVP and VIP neurons project both mono- and polysynaptically to the PVN where they release AVP and GABA to inhibit pre-autonomic PVN neurons. These PVN neurons project directly and indirectly (through the rostral ventrolateral medulla) to neurons in the IML that project to noradrenergic neurons in the stellate ganglion that innervate the heart. Other autonomic rhythms that are likely controlled by the same or similar SCN output circuitry include those of hepatic glucose production, blood pressure, and adipocyte thermogenesis [134,146,147,148].

## 6. Outputs from the SCN: Behavioral Rhythms

Sleep—probably the most obvious circadian behavior—is regulated by a “two-process model” in which a circadian drive for wakefulness opposes a homeostatic need for sleep [149,150]. While the neuronal mechanisms underlying sleep homeostasis are uncertain, the circadian circuitry regulating wakefulness has been at least partially mapped. In this circuit, the SCN projects both directly and indirectly (through the ventral, but not dorsal, SPZ) to the DMH [46,151,152,153]. GABAergic DMH projections inhibit the activity of sleep-promoting VLPO neurons while, simultaneously, glutamatergic DMH projections stimulate the activity of wake-promoting hypocretin neurons in the LH. This ultimately activates the ascending reticular activating system (comprising a cholinergic pathway from the brainstem and a monoaminergic pathway from the brainstem and caudal hypothalamus) which projects to cortex to promote and sustain wakefulness [154,155,156,157,158]. This multi-stage circadian output pathway allows for flexibility in the timing of sleep and can potentially explain at least some of the sleep timing differences seen in nocturnal and diurnal animals. 

Lesions of the DMH or the ventral (but not dorsal) SPZ also profoundly disrupt circadian rhythms in locomotor activity [152,159,160,161]. SCN VIP neurons are functionally connected to DMH neurons, as optogenetically stimulating VIP neurons inhibits firing in a subset of DMH-projecting SPZ neurons [51]. Consequently, the period and amplitude of locomotor activity rhythms are each shortened after SCN VIP neurons are genetically ablated [51,145]. Like most hypothalamic nuclei, the DMH comprises a heterogeneous population of neurons, but the subpopulation(s) of DMH neurons that regulate locomotor activity rhythms are not fully understood. Locomotor activity rhythms are unaffected by chemogenetically activating or genetically ablating GABAergic DMH neurons but are abolished when leptin receptor-expressing DMH neurons are genetically silenced [162,163]. Activating *mWake*-, brain-derived neurotrophic factor (BDNF)-, or prodynorphin-expressing DMH neurons each acutely increases locomotor activity. However, how these DMH neuron subtypes specifically affect locomotor activity rhythms is unknown [164,165,166]. Locomotor activity is also regulated by SCN projections to the ARC. Surgically disconnecting the SCN from the ARC (leaving both nuclei intact and functional) abolishes locomotor activity rhythms in constant darkness [167]. The specific cell types involved in the SCN-ARC circuit are unclear but likely include neuropeptide Y (NPY) and agouti-related peptide (AgRP)-co-expressing and kisspeptin-expressing neurons [168,169].

The DMH and ARC are also implicated in the circadian control of food intake. The DMH was once proposed to be the site of the so-called “food entrainable oscillator” responsible for the persistence of food anticipatory activity after lesioning the SCN. More recent studies have instead definitively concluded that the DMH only modulates food anticipatory activity which persists after complete DMH ablation [170,171,172]. However, the DMH is a critical component of the SCN-to-feeding behavior output pathway, as DMH lesions disrupt or completely abolish circadian feeding rhythms [152,173]. The specific DMH cell types involved in feeding rhythmicity are not fully understood. Chemogenetically inhibiting tropomyosin receptor kinase B (TrkB)-expressing DMH neurons increases food intake during the light (inactive) phase but not the dark (active) phase of a light:dark cycle [174]. Similarly, genetically silencing leptin receptor-expressing DMH neurons disrupts diurnal rhythms in food intake [162]. These neurons normally inhibit orexigenic NPY/AgRP co-expressing ARC neurons that are essential for the regulation of feeding behavior [175,176]. Circadian rhythms in food intake are abolished in mice with targeted chemical ablation of either NPY/AgRP neurons or leptin receptor-expressing ARC neurons, which include both NPY/AgRP neurons and α-MSH-expressing neurons (a subset of anorexigenic pro-opiomelanocortin (POMC)/cocaine and-amphetamine-regulated transcript (CART) co-expressing ARC neurons) [177,178]. Although the SCN projects strongly to the ARC, it is unclear if either or both POMC/CART or NPY/AgRP neurons are under direct circadian control because they normally work together to induce or suppress feeding behavior. Genetically silencing neurotransmission in kisspeptin-expressing ARC neurons also disrupts feeding rhythms, likely by disrupting their normal inhibition of NPY/AgRP and excitation of POMC/CART signaling [169]. Intriguingly, a light pulse during the subjective night inhibits feeding by activating SCN AVP neuron projections to oxytocin-producing neurons in the PVN [179]. It is unclear if this is solely a light-driven inhibition of feeding behavior (e.g., masking) or is another mechanism by which the SCN normally regulates feeding rhythms in the absence of mistimed light exposure. SCN AVP neurons also project directly to thirst-promoting neurons in the OVLT. Optogenetic activation of these AVPergic projections increases, and optogenetic inhibition suppresses, fluid intake, suggesting that this circuit regulates thirst to mediate drinking behavior [180].

While more “complex” behaviors are either demonstrably or likely to be rhythmic, the output pathways from the SCN to the brain regions regulating these behaviors are mostly unmapped. However, a polysynaptic circuit for one such behavioral rhythm, aggression, has been traced from the SCN, through the SPZ, to the VMH [181]. In this circuit, SCN VIP neurons project to GABAergic neurons in the dorsal SPZ that innervate neurons in the ventrolateral VMH. Aggressive behavior in mice normally peaks at the beginning of the dark (active) phase, but the phase of this rhythm shifts to the beginning of the light (inactive) phase when the vesicular GABA transporter (VGAT) is selectively deleted in SPZ neurons. Accordingly, chemogenetically inhibiting GABAergic SPZ neurons increases aggression at dawn but has no effect at dusk. Finally, several indirect projections from the SCN to brain regions that regulate arousal, motivation, and mood have been identified. For example, the SCN projects polysynaptically to the VTA via the DMH, LHb, and MPN, the locus coeruleus via the MPOA and DMH, the dorsal raphe nucleus via the DMH, and the nucleus accumbens via the VTA and PVT [46,182,183,184]. The specific cell types involved in these SCN output circuits and how they integrate SCN timing information to rhythmically modulate behavior have not yet been determined.

## 7. Outputs from the SCN: Non-Synaptic Signaling

Despite limited synaptic connections, the SCN manages to synchronize clocks throughout the brain and body. While some of this signaling is polysynaptic, the SCN has several ways to communicate non-synaptically with these downstream oscillators. As discussed above, the SCN sets the phase of the circadian glucocorticoid rhythm by regulating the activity of CRH and pre-autonomic neurons in the PVN [139]. This hormonal timing cue is released from the adrenal glands into the circulatory system and can bind to glucocorticoid receptors (GRs) expressed on most cells and tissues [87]. Within these cells, glucocorticoid-GR complexes interact with glucocorticoid response elements upstream of the *Per1* promoter to rapidly upregulate Per1 expression and consequently set the phase of the molecular clock [185,186,187]. The SCN is thus able to entrain daily glucocorticoid release which in turn entrains extra-SCN brain clocks and peripheral tissue clocks. This glucocorticoid-mediated entrainment may, in part, explain the phase differences in behavior and physiology observed in animals with different temporal niches. In both nocturnal and diurnal species, SCN activity peaks during the middle of the subjective day. Conversely, glucocorticoid rhythms peak just prior to waking (subjective dusk or subjective dawn) [144,188,189,190]. Another “systemic” cue that can synchronize clocks downstream from the SCN is the circadian rhythm in body temperature. Mammals undergo a daily rhythm in core body temperature that has a peak-trough amplitude of about 2 °C and peaks during the mid-to-late active phase [191]. Just as the GR-absent SCN is insensitive to entraining glucocorticoid signals, the SCN of adult rodents is also resistant to circadian variations in temperature. However, clocks in peripheral tissues including the liver, lungs, and pituitary gland are highly sensitive to, and thus can be entrained by, these daily temperature changes orchestrated by the SCN [192,193]. Increasing temperature in these tissues upregulates the interaction of heat shock factors with heat shock elements located on the promoter region of *Per2* to provide a timing cue to the molecular clock [194,195]. Importantly, intercellular coupling among neurons in the SCN neural network provides robustness against this weak entraining signal. The SCN indirectly generates these body temperature rhythms through its interactions with hypothalamic nuclei that regulate thermogenesis including the median preoptic nucleus, arcuate nucleus, and pre-autonomic neurons in the PVN [131,196,197,198]. 

Importantly, SCN neural projections to target brain regions are not necessary for all rhythmic outputs. Implanting a “donor” SCN into an arrhythmic SCN-lesioned animal restores circadian rhythms in behavior and physiology [199]. If the donor SCN is encapsulated in a fine mesh that prevents neuronal outgrowth, locomotor activity rhythms, but not endocrine or other physiological rhythms, are restored [200,201]. The SCN thus secretes diffusible substance(s) that can pass through the mesh capsule and entrain brain regions controlling locomotion. The intact SCN may normally release these “output factors” through synaptic or paracrine signaling via neural projections to target brain regions expressing the appropriate receptors. Additionally, the SCN may release these factors into the third ventricle where they can diffuse into the cerebrospinal fluid (CSF) to interact with receptors on target sites adjacent to the ventricles [202]. Finally, the SCN may release these factors into a recently discovered portal system that directly connects the vasculature of the SCN and the OVLT [203]. These diffusible output factors may include VIP, AVP, and GRP, as, for example, the paracrine secretion of these neuropeptides from a co-cultured wild-type SCN suffices to restore circadian gene expression rhythms in a circadian mutant SCN [204]. Similarly, the periodic application of exogenous AVP can drive firing rate rhythms in PVN cultures [27]. AVP levels are also rhythmic in the CSF of numerous species including non-human primates and rodents, but possibly not in humans [205,206,207,208,209]. Other candidate output factors include prokineticin 2 (PK2), transforming growth factor α (TGF-α), and cardiotrophin-like cytokine (CLC), which all transiently inhibit locomotor activity when administered intracerebroventricularly during the subjective night [22,23,24]. PK2 receptors are expressed in several SCN target areas including the SPZ, PVN, and DMH, suggesting local synaptic or paracrine PK2 release from SCN afferents [22,119]. Conversely, receptors for CLC are primarily located on neurons adjacent to the third ventricle, suggesting CLC is primarily released by the SCN into the CSF [23]. TGF-α seems to primarily interact with receptors on neurons both in the PVN and adjacent to the third ventricle [210]. How (and if) these various diffusible outputs interact to generate rhythms in behavior and physiology is unknown.

## 8. Conclusions

Clearly, the SCN “connectome” or “wiring diagram” is extremely complex. The SCN does not solely process environmental light cues from the retina as is typically depicted in a simplified eskinogram. Instead, it encodes synaptic inputs from dozens of discrete brain areas and hormonal cues from the periphery. Perhaps uniquely among brain circuits, the SCN also communicates with the rest of the brain and body via both (poly)synaptic projections throughout the nervous system and non-synaptic signals to distant cells, organs, and tissues. The SCN thus exists as a central nexus within a complex, dynamic circadian network that integrates information about the outside world (light intensity) with information about an animal’s internal state (arousal, motivation, hormone levels, etc.). This integrated circadian timing signal is propagated onward to synchronize and coordinate daily rhythms in behavior and physiology. Critically, these SCN target regions each have their own molecular clocks and, within the brain, “neuronal clocks”, or daily rhythms in action potential frequency [211]. In some cases, these local clocks are not endogenous or self-sustaining: when isolated from the SCN, these rhythms immediately or rapidly dampen. But in other cases, local clocks remain rhythmic for several days or even indefinitely without SCN input [212]. While ablating some of these local clocks disrupts behavioral and/or physiological rhythms (despite an intact SCN), ablating other local clocks has essentially no effect on rhythmicity [144,213,214]. Consequently, clocks within different SCN target regions have different strategies by which they encode circadian time to regulate their rhythmic outputs. The SCN connectome must therefore incorporate not only inputs to and outputs from the SCN, but also how these inputs and outputs influence, and are influenced by, local clocks. This holistic understanding of circadian rhythm generation in mammals is essential to determine how the disruption of these circuits can negatively affect human health.

## Figures and Tables

**Figure 1 biology-12-00508-f001:**
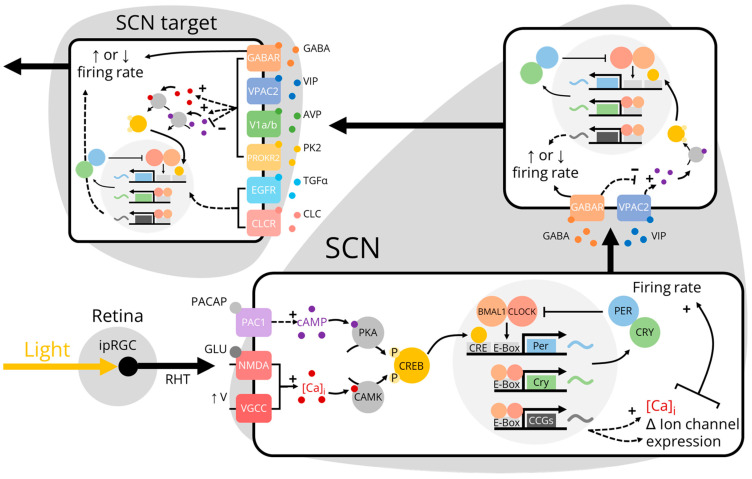
Photoentrainment of the SCN and SCN target neurons. Light activates intrinsically-photosensitive retinal ganglion cells (ipRGCs) which project through the retinohypothalamic tract (RHT) to release glutamate (GLU) and pituitary adenylate cyclase-activating peptide (PACAP) onto retinorecipient suprachiasmatic nucleus (SCN) neurons [9,10]. The activation of glutamate receptors increases intracellular calcium ([Ca]_i_) directly through N-methyl D-aspartate (NMDA) receptors and indirectly through voltage-gated calcium channels (VGCCs) [11]. The activation of PAC1 receptors initiates a G protein cascade that increases intracellular cyclic AMP (cAMP). cAMP and [Ca]_i_ activate protein kinase A (PKA) and calcium-dependent protein kinases (CAMK) to phosphorylate cAMP response element (CRE)-binding protein (CREB) [9,12]. Phosphorylated CREB promotes the transcription of the *Period* genes within the molecular circadian clock which increases the transcription of clock-controlled genes (CCGs) [13]. The translation of CCGs increases neuronal firing rate by directly increasing [Ca]_i_ or by altering membrane ion channel expression [14,15,16]. Increased firing rate in retinorecipient SCN neurons increases the release of gamma-aminobutyric acid (GABA) and vasoactive intestinal peptide (VIP) onto other neurons in the SCN neural network [17,18]. This changes neuronal firing rate and transcription of core clock genes either directly through the activation of ionotropic GABAA receptors or indirectly through metabotropic GABAB receptors and the VIP receptor VPAC2 [19,20,21]. SCN output neurons subsequently release several output factors including GABA, VIP, arginine vasopressin (AVP), prokineticin 2 (PK2), transforming growth factor-alpha (TGFα), and cardiotrophin-like cytokine (CLC) onto SCN target neurons [22,23,24,25,26,27]. These SCN output signals ultimately entrain SCN target neurons by changing their firing rates and core clock gene transcriptional activity.

**Figure 2 biology-12-00508-f002:**
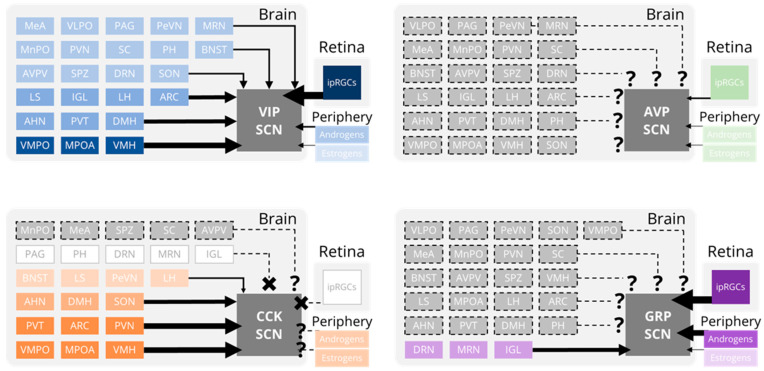
Inputs to the SCN. Schematics depicting the relative strengths of synaptic and hormonal inputs onto vasoactive intestinal peptide (VIP, blue), arginine vasopressin (AVP, green), cholecystokinin (CCK, orange), and gastrin-releasing peptide (GRP, purple) producing suprachiasmatic nucleus (SCN) neurons. Darker colors and thicker arrows represent stronger inputs from a given brain region or hormone. Gray boxes with no dashed outlines indicate brain regions with no inputs to an SCN subpopulation. Gray boxes with dashed outlines indicate brain regions whose (absence of) projections to an SCN subpopulation have not been experimentally determined. AHN, anterior hypothalamic nucleus; ARC, arcuate nucleus; AVPV, anteroventral periventricular nucleus; BNST, bed nucleus of the stria terminalis; DMH, dorsomedial hypothalamic nucleus; DRN, dorsal raphe nucleus; IGL, intergeniculate leaflet; ipRGCs, intrinsically-photosensitive retinal ganglion cells; LH, lateral hypothalamus; LS, lateral septum; MPOA, medial preoptic area; MRN, median raphe nucleus; MeA, medial amygdala; MnPO, median preoptic area; PAG, periaqueductal gray; PH, posterior hypothalamus; PVN, paraventricular nucleus of the hypothalamus; PVT, paraventricular nucleus of the thalamus; PeVN, periventricular nucleus; SC, superior colliculus; SON, supraoptic nucleus; SPZ, subparaventricular zone; VLPO, ventrolateral preoptic area; VMH, ventromedial hypothalamus; VMPO, ventromedial preoptic nucleus.

**Figure 3 biology-12-00508-f003:**
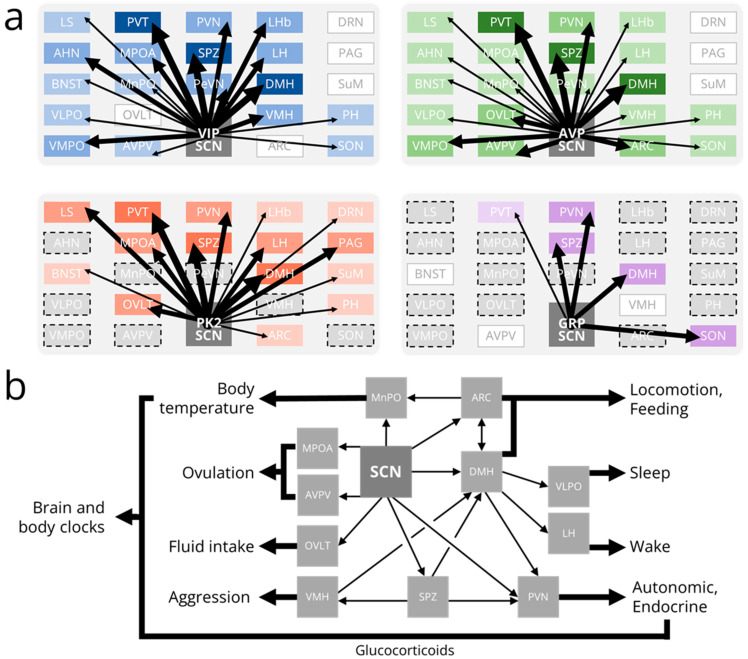
Outputs from the SCN. (**a**) Schematics depicting the relative strengths of synaptic outputs from vasoactive intestinal peptide (VIP, blue), arginine vasopressin (AVP, green), prokineticin 2 (PK2, red), and gastrin-releasing peptide (GRP, purple) producing suprachiasmatic nucleus (SCN) neurons. Boxes are roughly organized such that the brain region in the top left is the most rostral and most dorsal. Darker colors and thicker arrows represent stronger inputs onto a given brain region. Gray boxes with no dashed outlines indicate brain regions with no inputs from an SCN subpopulation. Gray boxes with dashed outlines indicate brain regions whose (absence of) inputs from an SCN subpopulation have not been experimentally determined. Note that the depicted GRP efferent projections have not been identified using modern genetic tools. (**b**) Schematic depicting output pathways from the SCN to several rhythmic behaviors and physiological processes. Glucocorticoid release and body temperature rhythms allow the SCN to synchronize distant brain and body clocks. AHN, anterior hypothalamic nucleus; ARC, arcuate nucleus; AVPV, anteroventral periventricular nucleus; BNST, bed nucleus of the stria terminalis; DMH, dorsomedial hypothalamus; DRN, dorsal raphe nucleus; LH, lateral hypothalamus; LHb, lateral habenula; LS, lateral septum; MPOA, medial preoptic area; MnPO, median preoptic area; OVLT, organum vasculosum of the lamina terminalis; PAG, periaqueductal gray; PH, posterior hypothalamus; PVN, paraventricular nucleus of the hypothalamus; PVT, paraventricular nucleus of the thalamus; PeVN, periventricular nucleus; SON, supraoptic nucleus; SPZ, subparaventricular zone; SuM, supramammillary nucleus; VLPO, ventrolateral preoptic nucleus; VMH, ventromedial hypothalamus; VMPO; ventromedial preoptic nucleus.

## Data Availability

Not applicable.

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
