# Peer review of "Inputs and Outputs of the Mammalian Circadian Clock"

_biology, 2023, doi:10.3390/biology12040508_

Round 1

Reviewer 1 Report

This review article attempts to explain how the mammalian circadian clock is entrained through the suprachiasmatic nucleus (SCN). While the authors have gathered a significant amount of information regarding the input and output of the SCN, the abundance of information can cause confusion and disrupt focus on the main concept. To improve clarity and understanding of the big picture of entrainment through the SCN, it would be beneficial to align the information with the signal transduction pathway of the circadian clock. This could be achieved through the use of a figure that shows the signal transduction pathway through the SCN and highlights the major signal molecules involved. The writing should provide a clearer view of the major pathway of entrainment. The variations between species could be explained based on the mainstream events in the pathway.

Author Response

  1. This review article attempts to explain how the mammalian circadian clock is entrained through the suprachiasmatic nucleus (SCN). While the authors have gathered a significant amount of information regarding the input and output of the SCN, the abundance of information can cause confusion and disrupt focus on the main concept. To improve clarity and understanding of the big picture of entrainment through the SCN, it would be beneficial to align the information with the signal transduction pathway of the circadian clock.

We thank the reviewer for their insightful comments. The focus of our review was not necessarily to focus on entrainment per se, but to synthesize the many disparate studies that describe the connectivity of the SCN: from where it receives information, and to where it sends this information once processed. That said, photoentrainment is a major input to the SCN and we agree that we should include a figure showing the “canonical” input-output pathway of the SCN (see below).

  • This could be achieved through the use of a figure that shows the signal transduction pathway through the SCN and highlights the major signal molecules involved. The writing should provide a clearer view of the major pathway of entrainment. The variations between species could be explained based on the mainstream events in the pathway.

We have included a new Figure 1 that demonstrates the “canonical” photoentrainment pathway from light, to the retina, to the SCN, to SCN targets.

Reviewer 2 Report

The manuscript by Starnes and Jones reviews current understanding of afferent and efferent connections from and to the suprachiasmatic nucleus (SCN), the site of the endogenous circadian oscillator.

The review is comprehensive but lacks at several places original literature, as the authors only refer to recent publications or reviews and do not cite original work.

An important  weakness of the manuscript is the neglect of the feedback loop of SCN-driven melatonin synthesis on the SCN itself via high affinity melatonin receptors, which has a major physiological function at least within maternal-foetal and post-partum communication. The authors should add a paragraph on this particular hormonal input on SCN function and circadian rhythm generation.

Author Response

  1. The manuscript by Starnes and Jones reviews current understanding of afferent and efferent connections from and to the suprachiasmatic nucleus (SCN), the site of the endogenous circadian oscillator.

  • The review is comprehensive but lacks at several places original literature, as the authors only refer to recent publications or reviews and do not cite original work.

We have updated our citations to include primary research articles in addition to review articles.

  • An important weakness of the manuscript is the neglect of the feedback loop of SCN-driven melatonin synthesis on the SCN itself via high affinity melatonin receptors, which has a major physiological function at least within maternal-foetal and post-partum communication. The authors should add a paragraph on this particular hormonal input on SCN function and circadian rhythm generation.

We have added a paragraph on melatonin input to the SCN in Section 3.

Reviewer 3 Report

Overview: 

The authors provide a well-written and comprehensive review of the anatomical inputs and outputs of the mammalian circadian clock. This review builds on and updates previous anatomical reviews of the suprachiasmatic nucleus. The analogous reviews which come to mind include "The circadian visual system, 2005" from L.P. Morin and C.N. Allen, as well as the follow-up review from L.P. Morin written in 2013 entitled "Neuroanatomy of the extended circadian rhythm system." From what I can gather, this topic has not been updated with a review since 2013, so this contribution will be welcomed by the field. Indeed, in the last ten years, intersectional genetics (the Cre-lox system, etc) has revolutionized neuroscience and given researchers the unprecedented opportunity to investigate subpopulations of neurons. Such subpopulation analysis has provided new insights to heterogeneous hypothalamic nuclei such as the suprachiasmatic nucleus. The authors include discussion of many of such studies in this review.

Major comments: 

The figures are clear and useful; however, I have do have a suggestion. The authors use gray boxes with no dashed outlines to indicate brain regions with no inputs to a SCN subpopulation and gray boxes with dashed outlines to indicate brain regions whose (absence of) projections to a SCN subpopulation have not been experimentally determined. Gray boxes with dashes works well to represent a lack of information, but to more clearly indicate a lack of input (and to distinguish between these two categories), perhaps it would be better to use a white box with a black outline (and black font) for this purpose. 

In order to be comprehensive, the authors may wish to include (a discussion, a mention, or simply a reference of) the following studies: 

- This study, which describes GABAergic input from the retina to the SCN.

- The work which shows the expression of melatonin receptors in / input to the SCN. For example, this study: https://doi.org/10.1111/jpi.12575, or as discussed in this review (and others): https://doi.org/10.1111/febs.16233. 

- This study, which describes a function for dopamine input to the SCN: https://doi.org/10.1016/j.cub.2019.11.029. 

Minor comments: 

- In the abstract, it is written that the SCN "sends an output signal," but perhaps it is better to say, "sends output signals."

- The authors mention "conventional retrograde tract-tracing methods" at two or three points throughout the manuscript. A brief description of what  this method is and how it differs from more modern methods may be beneficial in order to refresh readers who do not specialize in anatomy. 

- The first sentence of Section 5 uses the phrase "these SCN projections," referring to the prior section. Consider removing the word "these." Similar thought for the first sentence of section 7. 

- Consider splitting the first paragraph in Section 6, into two paragraphs-- the first for sleep and the second for locomotor activity (line 291). 

- Section 7 discusses the idea that the SCN may signal by releasing factors into the cerebrospinal fluid. You may be interested to know that vasopressin (and maybe other neuropeptides) show a rhythm in their concentration in the cerebrospinal fluid. For example, see this study from the group of Steven Reppert: https://doi.org/10.1016/0006-8993(83)91205-2. (In regard to whether to add such information about the rhythms of neuropeptides in the CSF, I leave this decision to the authors.) 

Author Response

  1. The authors provide a well-written and comprehensive review of the anatomical inputs and outputs of the mammalian circadian clock. This review builds on and updates previous anatomical reviews of the suprachiasmatic nucleus. The analogous reviews which come to mind include "The circadian visual system, 2005" from L.P. Morin and C.N. Allen, as well as the follow-up review from L.P. Morin written in 2013 entitled "Neuroanatomy of the extended circadian rhythm system." From what I can gather, this topic has not been updated with a review since 2013, so this contribution will be welcomed by the field. Indeed, in the last ten years, intersectional genetics (the Cre-lox system, etc) has revolutionized neuroscience and given researchers the unprecedented opportunity to investigate subpopulations of neurons. Such subpopulation analysis has provided new insights to heterogeneous hypothalamic nuclei such as the suprachiasmatic nucleus. The authors include discussion of many of such studies in this review.

  • The figures are clear and useful; however, I do have a suggestion. The authors use gray boxes with no dashed outlines to indicate brain regions with no inputs to a SCN subpopulation and gray boxes with dashed outlines to indicate brain regions whose (absence of) projections to a SCN subpopulation have not been experimentally determined. Gray boxes with dashes works well to represent a lack of information, but to more clearly indicate a lack of input (and to distinguish between these two categories), perhaps it would be better to use a white box with a black outline (and black font) for this purpose. 

We have changed “no inputs” from gray boxes, no dashed outline to white boxes with dark gray outline and dark gray text.

  • In order to be comprehensive, the authors may wish to include (a discussion, a mention, or simply a reference of) the following studies: 
    • This study, which describes GABAergic input from the retina to the SCN.

We have added “Surprisingly, a subset of ipRGCs release GABA onto the SCN to dampen the sensitivity of the circadian system to dim light” to Section 2 and changed the end of that paragraph to read “This suggests that the ratio of GABA to glutamate released from ipRGCs onto the SCN is much greater in nocturnal rodents than in diurnal rodents.  The functional significance of these inhibitory projections in diurnal rodents is unknown but may mechanistically explain differences in photoentrainment between species” as well as the suggested reference.

  • The work which shows the expression of melatonin receptors in / input to the SCN. For example, this study: https://doi.org/10.1111/jpi.12575, or as discussed in this review (and others): https://doi.org/10.1111/febs.16233.

We have added a paragraph on melatonin input to the SCN in Section 3.

  • This study, which describes a function for dopamine input to the SCN: https://doi.org/10.1016/j.cub.2019.11.029. 

We have added “Dopamine receptor 1a-expressing SCN neurons receive sparse monosynaptic projections from dopaminergic neurons of the ventral tegmental area (VTA). The release of dopamine onto the SCN has been functionally linked to photoentrainment and, intriguingly, weight gain associated with hedonic feeding.” to Section 2 as well as the suggested reference.

  • In the abstract, it is written that the SCN "sends an output signal," but perhaps it is better to say, "sends output signals."

We have changed this line to “sends output signals.”

  • The authors mention "conventional retrograde tract-tracing methods" at two or three points throughout the manuscript. A brief description of what  this method is and how it differs from more modern methods may be beneficial in order to refresh readers who do not specialize in anatomy. 

We have changed this sentence in Section 2 to read: “Several non-retinal inputs to the SCN of nocturnal animals have also been identified using “conventional” retrograde tract-tracing methods (in contrast to more modern viral vector-based and genetically-encoded tracing methods; for review, see (Saleeba et al. 2019)).”

  • The first sentence of Section 5 uses the phrase "these SCN projections," referring to the prior section. Consider removing the word "these." Similar thought for the first sentence of section 7. 

We have deleted “of these” in the first sentence of Section 5 and “these” in the first sentence of Section 7.

  • Consider splitting the first paragraph in Section 6, into two paragraphs-- the first for sleep and the second for locomotor activity (line 291). 

We have split this paragraph into two paragraphs.

  • Section 7 discusses the idea that the SCN may signal by releasing factors into the cerebrospinal fluid. You may be interested to know that vasopressin (and maybe other neuropeptides) show a rhythm in their concentration in the cerebrospinal fluid. For example, see this study from the group of Steven Reppert: https://doi.org/10.1016/0006-8993(83)91205-2. (In regard to whether to add such information about the rhythms of neuropeptides in the CSF, I leave this decision to the authors.) 

We have added the sentence “AVP levels are also rhythmic in the CSF of numerous species including non-human primates and rodents, but possibly not in humans.” to Section 7 as well as several references.

Reviewer 4 Report

The authors review the information about synaptic and non-synaptic inputs onto and output from the suprachiasmatic nucleus (SCN). The aim is to describe the connectivity from this circadian clock that regulates the circadian rhythms of the behaviors and physiological processes in the organism.

Suggesting minor changes:

The authors should include a list of abbreviations to read this review easily, particularly Figs. 1 and 2.

Author Response

  1. The authors review the information about synaptic and non-synaptic inputs onto and output from the suprachiasmatic nucleus (SCN). The aim is to describe the connectivity from this circadian clock that regulates the circadian rhythms of the behaviors and physiological processes in the organism. Suggesting minor changes:

  • The authors should include a list of abbreviations to read this review easily,  particularly Figs. 1 and 2.

We have added a list of abbreviations to the end of the manuscript.

Round 2

Reviewer 2 Report

Incorporated changes improve the quality of the manuscript in a way, that the paper is now acceptable for publication.